# Prevalence of Chronic Pruritus in Elderly Black and White Inpatients: A Comparative Population Study

**DOI:** 10.3390/jcm12155025

**Published:** 2023-07-31

**Authors:** Omar Mahmoud, Siri Choragudi, Amanda Nwaopara, Gil Yosipovitch

**Affiliations:** Dr. Phillip Frost Department of Dermatology and Cutaneous Surgery, Miami Itch Center, Miller School of Medicine, University of Miami, Miami, FL 33146, USA; oxm219@med.miami.edu (O.M.); sxc1164@med.miami.edu (S.C.); aun1@med.miami.edu (A.N.)

**Keywords:** chronic, pruritus, itch, elderly, geriatric, Black, African American, White, inpatient, population, study

## Abstract

Background: Black and geriatric patients were reported in small scale studies to have more intense chronic pruritus (CP). Studies comparing itch across geriatric racial groups are lacking. Objectives: To compare the prevalence of CP in Black and White inpatients ≥ 65 years old as well as the top primary diagnoses of these populations. Methods: We used data from the National Inpatient Sample from 2016–2019 to analyze CP prevalence and ICD10-CM to identify diseases. The top five primary diagnoses were calculated for a subpopulation with CP. Sample characteristics were described, and the data was pooled and analyzed using IBM SPSS^®^ Complex Sample modules. Results: Among hospitalized Black inpatients ≥ 65 years old, the prevalence of CP was 0.26% while in the White cohort it was 0.22%. The top five primary diagnoses in the Black population with itch were sepsis (4.2%); hypertensive heart and chronic kidney disease (CKD) with heart failure (HF) and stage 1–4 CKD, or unspecified CKD (4.1%); acute kidney failure (4.0%); hypertensive heart and CKD with HF with stage 5 CKD, or end-stage renal disease (2.1%); and hypertensive heart disease with HF (1.7%). The top five primary diagnoses in the White population were sepsis (4.25%); acute kidney failure (3.0%); hypertensive heart and CKD with HF and stage 1–4 CKD, or unspecified CKD (2.5%); cellulitis of left lower limb (1.9%); and unilateral primary osteoarthritis, right knee (1.9%). Conclusions: Geriatric hospitalized Black patients demonstrated a higher prevalence of chronic itch compared with the White cohort, which may be related to the higher prevalence of chronic kidney disease in different stages of severity in this population.

## 1. Introduction

Chronic pruritus, defined as itching lasting for 6 weeks or more, imposes a substantial burden on affected patients [1]. The sensation of itch and urge to scratch can be debilitating, and patients suffering with chronic pruritus often experience limitations to their daily activities, function, and sleep, increased mood disorders including anxiety and depression, and a decline in their overall quality of life [2,3]. Chronic pruritus affects certain ethnoracial groups disproportionately. Previous small-scale epidemiologic studies have demonstrated a higher prevalence of chronic pruritus in Black individuals when compared with other ethnoracial groups [4]. Whang et al. reported in a cross sectional study of 18,753 patients at Johns Hopkins Health System that patients seen for itch-related concerns were more likely to be Black (37%) compared to White (19%), and noted that these findings correlated with findings of larger national databases [5]. A multi-center cross-sectional study reported similar findings of a higher prevalence of chronic pruritus in Black individuals compared with other racial groups [6].

There is also data to suggest that Black individuals experience a significantly higher itch intensity compared with other ethnoracial groups. A retrospective analysis of electronic health records of chronic pruritus patients at Temple University found that Black patients experienced the greatest itch intensity of the racial groups [7]. Likewise, elderly individuals were reported to experience greater itch intensity in comparison with younger individuals [7]. Similar to Black individuals, previous epidemiologic studies have demonstrated a higher prevalence of chronic pruritus in elderly individuals [8].

With the higher burden of itch in Black individuals, this prompts the need for larger population studies exploring the prevalence and characteristics of itch in this group. Currently, larger population-based analyses are limited, and there are no studies comparing chronic pruritus in Black elderly patients with other elderly ethnoracial groups.

## 2. Objectives

To compare the prevalence of chronic pruritus (ICD10-CM: L29.8 and L29.9) in Black and White inpatients aged 65 and older, as well as the top primary diagnoses in Black and White inpatients aged 65 and older with chronic pruritus.

## 3. Methods

We utilized the National Inpatient Sample (NIS) from 2016 to 2019, provided by the Healthcare Cost and Utilization Project (HCUP). The NIS sample represents 20% of hospitalization in the US and allows the calculation of national-level estimates of all hospital discharges across the US. We included Black adults ≥ 65 years older admitted to the hospitals from 2016 to 2019. We used the International Classification of Diseases, Tenth Revision, and Clinical Modification codes (ICD10-CM) for disease identification. We calculated the prevalence of chronic pruritus (ICD10-CM: L29.8 and L29.9). Furthermore, we selected a subpopulation of the sample with chronic pruritus to calculate the top five primary diagnoses (primary admission diagnosis). We summarized the sample using age (years), sex (male or female), race (White, Black, Hispanic and other), mean household income (<$39,000, $39,000–47,999, $48,000–62,999, $63,000 or more), and primary payer (Medicare, Medicaid, private insurance, self-pay, and no charge or other). We described the sample characteristics using survey-weighted means and 95% confidence intervals (CI) for continuous variables and unweighted and survey-weighted frequencies and prevalence for categorical variables. We pooled the data from 2016 to 2019 and used sample strata accounting for hospital characteristics and year of the NIS data, clusters, and discharge weights to calculate the national level estimates. Statistical analysis was conducted using complex sample modules of IBM SPSS^®^ Statistics 25.0 (IBM, Chicago, IL, USA), accounting for the complex sample design of the NIS. 

## 4. Results

Of 28,484,087 (weighted: 142,420,378) admissions from 2016 to 2019, we included 1,066,040 (weighted: 5,330,201) Black inpatients 65 years or older and 7,752,048 (weighted: 38,760,224) White inpatients 65 years or older (Table 1 and Table 2). The prevalence of chronic pruritus among Black inpatients aged 65 or more was 0.26% (95% CI 0.25–0.28%), and the top five primary diagnoses among this population were sepsis, unspecified organism (ICD10-CM: A41.9); hypertensive heart and chronic kidney disease with heart failure and stage 1 through stage 4 chronic kidney disease, or unspecified chronic kidney disease (ICD10-CM: I13.0); acute kidney failure, unspecified (ICD10-CM: N17.9); hypertensive heart and chronic kidney disease with heart failure and with stage 5 chronic kidney disease, or end-stage renal disease (ICD10-CM: I13.2); and hypertensive heart disease with heart failure (ICD10-CM: I11.0), with a prevalence of 4.2%, 4.1%, 4.0%, 2.1% and 1.7%, respectively. The prevalence of chronic pruritus among White inpatients aged 65 or more was 0.22% (95% CI 0.22–0.23%), and the top five primary diagnoses among this population were sepsis, unspecified organism (ICD10-CM: A41.9); acute kidney failure, unspecified (ICD10-CM: N17.9); hypertensive heart and chronic kidney disease with heart failure and stage 1 through stage 4 chronic kidney disease, or unspecified chronic kidney disease (ICD10-CM: I13.0); cellulitis of the left lower limb (ICD10-CM: L03.116); and unilateral primary osteoarthritis, right knee (ICD10-CM: M17.11), with a prevalence of 4.25%, 3.0%, 2.5%, 1.9%, and 1.9%, respectively.

## 5. Discussion

In this study, we compared the prevalence of chronic pruritus between Black patients ≥ 65 years old and White patients ≥ 65 years old in the inpatient setting using data from the National Inpatient Sample from 2016 to 2019, and correlated these findings with the top primary diagnoses and sociodemographic characteristics of these populations. Geriatric hospitalized Black patients demonstrated a higher prevalence of chronic itch at 0.26% (95% CI 0.25–0.28%), compared with the White cohort at 0.22% (95% CI 0.22–0.23%). Due to the large sample sizes of this study, we note these differences to be significant.

The higher prevalence in the Black cohort may be due to the higher prevalence of chronic kidney disease (CKD) in this population. The prevalence of hypertensive heart and CKD with heart failure (HF) and stage 1–4 CKD, or unspecified CKD is 4.1% in the Black cohort compared with 2.5% in the White cohort. Additionally, hypertensive heart and CKD with HF with stage 5 CKD, or end-stage renal disease was reported as the fourth highest prevalence of the top five primary diagnoses in the Black cohort at 2.1%, whereas, this diagnosis was not in the top five primary diagnoses in the White cohort. Itch is a very common symptom of CKD, and presents as a systemic itch that most commonly affects the back, but also the head, abdomen, and arms [9]. Prevalence of acute kidney failure was higher in the Black cohort (4.0%) compared with the White cohort (3.0%), although this difference is less likely than CKD to contribute to the higher prevalence of chronic pruritus seen in the Black cohort. Sepsis was the top primary diagnosis in both cohorts with a similar prevalence (4.2% in the Black cohort and 4.25% in the White cohort), although this is also not likely to contribute to the itch. The Black cohort demonstrated a lower mean household income and a higher prevalence of Medicaid patients; however, we do not believe this could be interpreted to be related to itch, as with the other sociodemographic differences between the cohorts.

The higher prevalence of itch in the Black cohort could be due ethnoracial differences in the prevalence of chronic pruritic disorders such as prurigo nodularis (PN) and atopic dermatitis (AD). Data from the Johns Hopkins Hospital System reported that Black patients were 3.4 times more likely to have PN than White patients [10]. AD prevalence is also reported to be higher in Black individuals compared with other ethnoracial groups. A large scale study of over 100,000 children demonstrated that Black children have significantly higher rates of AD than Caucasian children (15.9% vs. 9.7%) [11]. Another study found that Black patients were 50% more likely to have ambulatory visits for AD than White patients [12]. The higher prevalence of these chronic pruritic conditions could explain the differences in itch we see in our population study. However, in the inpatient setting we did not find data on AD.

Differences in Black skin structure and physiology may contribute to itch. Black patients have a lower stratum corneum ceramide content compared with other racial groups which contributes to increased transepidermal water loss, skin dryness, and the activation of itch-associated nerve fibers that induce itch sensation [13]. Other structural differences demonstrated in Black skin include a lower stratum corneum pH and increased mast cell size in the stratum corneum, as well as genetic polymorphisms in mas-related G protein coupled receptors, transient receptor potential vanilloid receptors, and filaggrin [4]. The extent to which these structural and physiologic differences contribute to population-level differences in chronic pruritus remains to be explored.

The findings of our study may underestimate the prevalence of chronic pruritus due to the underreporting of itch by patients. Additionally, in Black skin, especially elderly Black skin, the clinical presentation and skin signs of AD are not classical, which may potentially lead to underdiagnosis of chronic pruritus [1]. In addition to chronic diseases, dry skin is more common in the elderly, contributing to itch. In future studies, we aim to explore various etiologies of chronic itch at the population level, encompassing dermatologic, systemic, neuropathic, and psychogenic causes, as well as medications, blood abnormalities, dry skin, and malignancy [14]. Additionally, investigating differences in itch intensity and characteristics, such as distribution and quality, among specific chronic pruritus conditions across different ethnoracial groups, is warranted. These findings will contribute to a deeper understanding of itch in the Black population and aid us in addressing this unmet need more effectively.

## Figures and Tables

**Table 1 jcm-12-05025-t001:** Descriptive statistics of the sample—elderly Black and White populations NIS 2016–2019 (NIS: National inpatient sample; CI: confidence interval).

	Black Cohort		White Cohort	
Variables	Weighted Frequency (Unweighted Frequency) 5,330,201 (1,066,040)	Prevalence (%) (95% CI)	Weighted Frequency (Unweighted Frequency) 38,760,224 (7,752,048)	Prevalence (%) (95% CI)
Age (years), mean (95% CI)	5,330,201 (1,066,040)	75.31 (75.27–75.34)	38,760,224 (7,752,048)	77.25 (77.22–77.28)
Sex				
Male	2,266,265 (453,253)	42.5 (42.4–42.7)	17,941,358 (3,588,273)	46.3 (46.2–46.4)
Female	3,062,900 (612,580)	57.5 (57.3–57.6)	20,818,866 (4,163,775)	53.7 (53.6–53.8)
Mean household income				
<USD 39,000	2,746,880 (549,376)	52.3 (51.4–53.2)	9,141,868 (1,828,375)	23.6 (23.2–24.0)
USD 39,000–$47,999	1,095,370 (219,074)	20.9 (20.4–21.4)	10,653,574 (2,130,716)	27.5 (27.1–27.9)
USD 48,000–USD 62,999	840,315 (168,063)	16.0 (15.6–16.5)	10,116,437 (2,023,288)	26.1 (25.7–26.5)
≥USD 63,000	568,525 (113,705)	10.8 (10.4–11.3)	8,848,344 (1,769,669)	22.8 (22.3–23.4)
Primary expected payer				
Medicare	4,614,870 (922,974)	86.7 (86.3–87.1)	35,061,840 (7,012,371)	90.5 (90.3–90.6)
Medicaid	141,205 (28,241)	2.7 (2.5–2.8)	200,220 (40,044)	0.5 (0.5–0.6)
Private insurance	435,490 (87,098)	8.2 (7.8–8.5)	2,761,490 (552,298)	7.1 (6.9–7.3)
Self-Pay	36,210 (7,242)	0.7 (0.6–0.7)	134,410 (26,882)	0.3 (0.3–0.4)
No charge or other	95,725 (19,145)	1.8 (1.7–1.9)	602,265 (120,453)	1.6 (1.5–1.6)
Pruritus (ICD10-CM: L29.8 and L29.9)—primary and secondary diagnoses				
Yes	14,100 (2,820)	0.26 (0.25–0.28)	86,595 (17,319)	0.22 (0.22–0.23)
No	5,316,100 (1,063,220)	99.74 (99.72–99.75)	38,673,629 (7,734,729)	99.78 (99.77–99.78)

**Table 2 jcm-12-05025-t002:** Top 5 primary diagnoses among inpatients with pruritus (ICD 10: L29.8 and L29.9)—elderly Black and White populations NIS 2016–2019 (NIS: National inpatient sample; CI: confidence interval).

	Top 5 Primary Diagnoses among Inpatients with Pruritus (ICD 10: L29.8 and L29.9)	Weighted Frequency (Unweighted Frequency)	Prevalence (%) (95% CI)
Black Cohort	Sepsis, unspecified organism (ICD10-CM: A41.9)	595 (119)	4.2 (3.6–5.0)
	Hypertensive heart and chronic kidney disease with heart failure and stage 1 through stage 4 chronic kidney disease, or unspecified chronic kidney disease (ICD10-CM: I13.0)	575 (115)	4.1 (3.4–4.9)
	Acute kidney failure, unspecified (ICD10-CM: N17.9)	570 (114)	4.0 (3.4–4.8)
	Hypertensive heart and chronic kidney disease with heart failure and with stage 5 chronic kidney disease, or end stage renal disease (ICD10-CM: I13.2)	300 (60)	2.1 (1.7–2.7)
	Hypertensive heart disease with heart failure (ICD10-CM: I11.0)	235 (47)	1.7 (1.2–2.2)
White Cohort	Sepsis, unspecified organism (ICD10-CM: A41.9)	3930 (786)	4.25 (4.2–4.8)
	Acute kidney failure, unspecified (ICD10-CM: N17.9)	2665 (533)	3.0 (2.8–3.3)
	Hypertensive heart and chronic kidney disease with heart failure and stage 1 through stage 4 chronic kidney disease, or unspecified chronic kidney disease (ICD10-CM: I13.0)	2235 (447)	2.5 (2.3–2.8)
	Cellulitis of left lower limb (ICD10-CM: L03.116)	1705 (341)	1.9 (1.7–2.2)
	Unilateral primary osteoarthritis, right knee (ICD10-CM: M17.11)	1700 (340)	1.9 (1.6–2.3)

## Data Availability

Not applicable.

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
