# Peer review of "Prevalence of Chronic Pruritus in Elderly Black and White Inpatients: A Comparative Population Study"

_jcm, 2023, doi:10.3390/jcm12155025_

Round 1

Reviewer 1 Report

The authors aimed to characterize the incidence and etiology of chronic pruritus in black elderly inpatient patients in a large population compared to the white counterpart.

A few comments:

1)     In the introduction/discussion– please note other etiologies of chronic pruritus which are not included in the study's findings e.g.- medications, blood abnormalities such as anemia, xerotic dermatitis, and malignancy and why these etiologies were not found in this cohort.

2)     Lines 102-104 "Itch is a very 102 common symptom of CKD, and presents as a systemic itch that most commonly affects 103 the back, but also the head, abdomen, and arms", please state your reference

1)    Line 88 duplication of the word "compared"

2)     Line 120 "liketly"

Reviewer 2 Report

Dear authors 

The number of patients in your study is quite good.

However, in addition to chronic diseases, dry skin may be common in elderly patients.

I suggest you make additional evaluations on this subject in the future. In addition, drugs (such as anti-hypertensive drugs) can cause itching by having side effects.

Reviewer 3 Report

I appreciate the effort you have put into your research and the chance to provide feedback to improve the quality of your work. 

First, I would like to acknowledge the main strenght of your manuscript, the precision of the results of your study due to the sample size and that all data are real world data.

Yet after careful consideration, I also have the following concerns:

1. The prevalence of itch is different in elderly black and white inpatients; however, I suggest to analize in the Discussion section if this difference in prevalence is really a minimally important and clinical difference.

2. It is important to explain why despite the higher prevalence of CKD among black patients, the prevalence of itch did not increase in the same way. I suggest to explore if there is a correlation between the prevalence of itch and the duration of the CKD if you have the data; and also with the days of hospitalization (skin care during hospitalization) and drugs administered that could increase the perception of itch. 

3. I suggest to combine Table 1 and 2 in one table.

In conclusion, I believe your manuscript has the potential to make a valuable contribution to the field. However, I strongly recommend that you address the above major comments before resubmitting yur manuscript for further review.

Thank you for the opportunity to review your work.

Round 2

Reviewer 3 Report

All the recommendations were addressed.